# Interhemispheric Anti-Phase Variability in Mesospheric Climate Driven by Summer Polar Upwelling During Solstice Months

Liang Zhang<sup>1</sup>, Zhongfang Liu<sup>1</sup>, Brian Tinsley<sup>2</sup>

State Key Laboratory of Marine Geology, Tongji University, Shanghai, 200092, China Physics Department, University of Texas at Dallas, Richardson, Texas, 75080, USA

Correspondence to: Liang Zhang (Liangzhang420@tongji.edu.cn)

Abstract. The upper mesosphere, a transition region between Earth's atmosphere and space, is characterized by complex interactions among water vapor (H<sub>2</sub>O), atomic hydrogen (H), ozone (O<sub>3</sub>), atomic oxygen (O), and temperatures. Using MLS, SABER, and SOFIE satellite data, we explore the upwelling-driven interannual variability of temperatures near 90 km (T90) and atmospheric constituents during solstice months, revealing a bottom-up control mechanism of "upwelling—H<sub>2</sub>O(H)—O<sub>3</sub>(O)—T90" in the two hemispheres. First, summer polar upwelling transports H<sub>2</sub>O upward, which is then transported toward the winter hemisphere by meridional winds. Subsequently, the hydration increases H via photolysis and depletes O<sub>3</sub> in the winter hemisphere through H-driven catalytic loss. The O varies in pace with O<sub>3</sub> due to chemical equilibrium, and the radiative and chemical heating by O/O<sub>3</sub> reactions reduces the T90 in the winter hemisphere (T90<sub>w</sub>). Second, upwelling-induced cooling promotes polar mesospheric cloud (PMC) formation, with ice particle growth blocking H<sub>2</sub>O transport and dehydrating heights above PMCs. This dehydration reduces H abundance, thereby decreasing H-driven O<sub>3</sub> loss. Meanwhile, the colder temperatures directly increase O<sub>3</sub> through ozone kinetics. The enhanced O<sub>3</sub>, together with the coupled O, collectively increase the summer polar temperatures above 90 km (T90<sub>s</sub>). This anti-phase interannual variability between hemispheres, mediated by PMC microphysics and H<sub>2</sub>O-O<sub>3</sub> chemistry, establishes summer polar upwelling as a fundamental driver of mesospheric climate and highlights the importance of dynamical-chemical coupling in the upper mesosphere.

## 1 Introduction

The summer upper mesosphere is the coldest region on Earth's atmosphere, with temperatures as low as 130 K due to the adiabatic cooling of gravity wave-driven upwelling (Plane *et al.*, 2023). During solstice months, zonal winds are westward below ~90 km and eastward above, and meridional winds flow from summer to winter hemispheres (Ramesh *et al.*, 2024). Despite recent advances in mesospheric wind observations, direct measurements of vertical winds remain challenging (Lee *et al.*, 2024; Vincent *et al.*, 2019; Zhang *et al.*, 2020). The upper mesosphere exhibits complex couplings between dynamics and chemistry, particularly through H<sub>2</sub>O and O<sub>3</sub> interactions that remain incompletely understood.

## 30 1.1 Water vapor and PMC dynamics

Upwelling and methane oxidation are primary H<sub>2</sub>O sources in the upper mesosphere (Lübken et al., 2018; Shi et al., 2023). The rapid decrease of H<sub>2</sub>O with altitude in the upper mesosphere (Figure 1a) is mainly caused by the photolysis of solar ultraviolet (UV) radiation, particularly in the summer polar region under continuous illumination. Fig. 1 demonstrates how summer polar upwelling creates concurrently temperature minima and H<sub>2</sub>O maxima in the summer hemisphere. These conditions promote PMC formation between 80 km and 90 km (Rapp and Thomas, 2006), which subsequently redistributes H<sub>2</sub>O through the well-established freeze-drying effect (Hervig et al., 2015; von Zahn and Berger, 2003). This top-down process, involving ice particle nucleation, growth, sedimentation, and sublimation, produces dehydration above PMCs and subsequently hydration below (Hultgren and Gumbel, 2014), with clear H<sub>2</sub>O depletion visible above 60°S-80°S PMCs in Fig. 1a. Siskind et al. (2018) successfully simulated the dehydration and hydration above and below PMCs, which extend to midlatitudes due to meridional circulations (Fig. 1 therein). Current PMC models systematically overestimate the dehydration/hydration magnitudes (Bardeen et al., 2010; Lübken et al., 2009), suggesting an incomplete understanding of PMC formation. PMC variability is modulated by various dynamical processes, including gravity waves (Gao et al., 2018), planetary waves (Liu et al., 2015), tides (Fiedler and Baumgarten, 2018), and inter-hemispheric coupling (Gumbel and Karlsson, 2011). While solar UV theoretically influences H<sub>2</sub>O and thus PMCs (Remsberg et al., 2018), the 11-year solar signal is insignificant over the past two decades (Hervig et al., 2019; Vellalassery et al., 2023), highlighting the need to identify alternative drivers.

#### 1.2 Ozone chemistry and variability

In the upper mesosphere, the O concentrations increase sharply with altitude due to solar photolysis of  $O_2$  (Mlynczak *et al.*, 2013), and the three-body recombination reaction of O produces a secondary ozone layer near 90 km (O+O<sub>2</sub>+M $\rightarrow$ O<sub>3</sub>+M) (Smith *et al.*, 2013). The O exhibits a long photochemical lifetime near the mesopause (days to months), consequently, its daily and seasonal variability is governed by vertical transport processes, including tides and downwelling of the mean circulation (Smith *et al.*, 2010b). O<sub>3</sub> is primarily destroyed by photolysis (O<sub>3</sub>+hv $\rightarrow$ O( $^1$ D, $^3$ P)+O<sub>2</sub>) during daylight with  $\sim$ 2 minute lifetime, and by reaction with hydrogen (H+O<sub>3</sub> $\rightarrow$ OH+O<sub>2</sub>) and oxygen (O+O<sub>3</sub> $\rightarrow$ 2O<sub>2</sub>) at night (Huang *et al.*, 2008; Smith and Marsh, 2005). The ozone photochemical equilibrium assumption, i.e., ozone loss is balanced by ozone production, is crucial for retrieving O and H concentrations (Kulikov et al., 2018; Mlynczak et al., 2007). The secondary ozone layer exhibits strong seasonal dependence, with peak O<sub>3</sub> mixing ratios occurring in winter (Fig. 1b) when reduced sunlight minimizes photolysis and lower H<sub>2</sub>O limits H availability. Temperature further modulates ozone kinetics, with colder conditions enhancing O<sub>3</sub> by accelerating production rate and inhibiting loss rates (Smith *et al.*, 2018).

Despite these advances in understanding ozone chemistry, key questions remain regarding ozone variability drivers. Solar activity appears to suppress ozone via production of odd hydrogen and odd nitrogen species (Jia *et al.*, 2024), with solar cycle signals and long-term trends occasionally documented (Huang *et al.*, 2014; Lee and Wu, 2020). PMCs may

enhance ozone through dehydration, which is expected to reduce the H above PMCs (Siskind *et al.*, 2008; Siskind *et al.*, 2018; Siskind and Stevens, 2006). The secondary ozone layer impacts mesospheric energy budgets through competing processes, including solar radiative heating, chemical heating, and infrared cooling (Ramesh *et al.*, 2015), and the total (radiative and chemical) heating of O and O<sub>3</sub> is of great importance for the energy budget of upper mesosphere (Mlynczak et al., 2018; Mlynczak and Solomon, 1993).

### 1.3 Temperature trends and variability



Mesospheric temperatures display complex variability patterns across multiple timescales. Interannual variations (4~5 K), outweigh the 11-year solar signal (1~3 K/100 solar flux unit) and the long-term cooling trend (-1~-2 K/decade) (French *et al.*, 2020a; French *et al.*, 2020b). In the summer polar mesopause, solar signals are unexpectedly absent in temperature, presumably due to a compensating dynamical cooling effect (Karlsson and Kuilman, 2018; Qian *et al.*, 2019). A well-established CO<sub>2</sub>-induced cooling trend prevails throughout most of the middle atmosphere, with a rate exceeding tropospheric warming magnitudes (Feofilov and Kutepov, 2012; Laštovička, 2017; Roble and Dickinson, 1989). This strong cooling is regarded as a potential indicator of climate change (Liu *et al.*, 2024; Solomon *et al.*, 2018). However, the summer polar mesopause presents a notable exception, exhibiting an unexpected warming trend that contradicts the dominant CO<sub>2</sub> cooling. Proposed explanations include shrinking effect (Bailey *et al.*, 2021; Dawkins *et al.*, 2023; Lübken *et al.*, 2013) and long-period vertical oscillations of temperature profile (Kalicinsky *et al.*, 2018; Offermann *et al.*, 2021), though the primary mechanism remains elusive. Stratospheric ozone recovery further complicates this picture by modulating gravity wave propagation, which subsequently affects mesospheric winds and temperatures (Smith *et al.*, 2010a; Venkateswara Rao *et al.*, 2015).

These interconnected physical and chemical processes create substantial challenges for a comprehensive understanding of the climate in upper mesosphere. This paper elucidates a systematic bottom-up mechanism, linking summer polar upwelling to interannual variability in H<sub>2</sub>O, H, O<sub>3</sub>, O, and temperatures. Section 2 details the satellite datasets (MLS, SABER, SOFIE) and theoretical framework, Section 3 quantifies the hemispheric climate patterns during solstice months (December/June), Section 4 discusses variability mechanism, and Section 5 summarizes key conclusions.

**Figure 1.** December climatology of (a) water vapor mixing ratio, (b) ozone mixing ration, and (c) temperature in the upper mesosphere. Data represent monthly means from MLS/Aura observations during 2004-2022.

## 90 2 Data and method





## 2.1 Multi-satellite Data

This study utilizes measurements from three satellite instruments to investigate upper mesospheric climate variability. The Microwave Limb Sounder (MLS) aboard NASA's Aura satellite, launched in July 2004, provides global atmospheric measurements between 82°N and 82°S (Jiang et al., 2007). Operating in a sun-synchronous orbit with equatorial crossings at 01:30 (ascending) and 13:30 (descending) local time, MLS delivers vertical profiles of H<sub>2</sub>O (0.001 hPa top level recommended, 7~9 km vertical resolution, 30% precision), O<sub>3</sub> (0.001 hPa, 5~7 km, 35%), and temperature (0.00046 hPa, 6~12 km, 3 K). Daily mean values were generated by averaging ascending and descending orbits, focusing on zonal (5° gridded) and monthly means for interannual variability studies. Fig. 1 shows the December climatology of H<sub>2</sub>O, O<sub>3</sub>, and temperature observed by MLS.

The Sounding of the Atmosphere using Broadband Emission Radiometry (SABER) instrument aboard the Thermosphere, Ionosphere, Mesosphere Energetics, and Dynamics (TIMED) satellite was launched in December 2001. With an orbital inclination of 74.1°, SABER's limb scan provides latitude coverage alternating between 83°N to 52°S (during June) and 52°N to 83°S (during December) due to its 60-day yaw cycle (Russell et al., 1999). This 60-day yaw cycle allows interannual comparison by maintaining stable latitude and local time coverage for a given month across different years. We used TIMED/SABER version 2.07 data between 0.01 and 0.003 hPa from 2002 to 2019, including temperature (Remsberg et al., 2008), ozone (Rong et al., 2009; Smith et al., 2013), atomic oxygen (Mlynczak et al., 2013; Mlynczak et al., 2018), and atomic hydrogen (Mlynczak et al., 2014).

The Solar Occultation for Ice Experiment (SOFIE) instrument onboard the Aeronomy of Ice in the Mesosphere (AIM) satellite was launched on 25 April 2007 into a sun-synchronous polar orbit (Russell III *et al.*, 2009). SOFIE covers latitudes between 65° and 82°, with particular focus on ~70° latitude during PMC seasons. Using solar occultation measurements, SOFIE obtains vertical profiles of PMC properties, temperature, water vapor, and ozone (Gordley *et al.*, 2009; Hervig *et al.*, 2009). SOFIE provides datasets for 8 PMC seasons (from 2007 to 2014) in the northern hemisphere (NH) and 7 PMC seasons (from 2007/2008 to 2013/2014) in the southern hemisphere (SH), after which SOFIE measurements shifted to lower latitudes where PMCs typically do not form.

#### 115 2.2 Bottom-up control mechanism framework

Our methodology examines the responses of mesospheric variables ( $H_2O$ , H,  $O_3$ , O, T90) to upwelling on interannual timescales. Since direct observations of summer polar upwelling are unavailable, we use temperature at ~80 km ( $T80_s$ ) as a proxy, based on its relationship with adiabatic cooling. Figure 2 illustrates the conceptual framework for how summer polar upwelling drives interannual climate variability during solstice months through the interconnected pathways in the "upwelling— $H_2O(H)$ — $O_3(O)$ —T90" chain:

(1) H<sub>2</sub>O/H variability. Upwelling produces both hydration that occurs below PMCs through direct transport of H<sub>2</sub>O by upwelling, and dehydration above PMCs through a "cold-trap effect". Explicitly, adiabatic cooling from upwelling lowers T80<sub>S</sub> promotes water vapor condensation into ice particles, thereby reduces upward H<sub>2</sub>O transport and causes dehydration above PMCs (Zhang *et al.*, 2025a). While similar to the conventional freeze-drying effect, the cold-trap effect shows distinct characteristics that we discuss later in section 4.1. Meridional winds then transport both the hydration and dehydration toward the winter hemisphere. Atomic hydrogen H, produced through H<sub>2</sub>O photolysis, consequently varies in synchrony with H<sub>2</sub>O in both hemispheres.




- (2) O<sub>3</sub>/O responses. O<sub>3</sub> abundance is negatively influenced by H<sub>2</sub>O through H acting as a primary O<sub>3</sub> sink (Zhang *et al.*, 2025b). Upwelling therefore enhances summer-hemisphere O<sub>3</sub> through dehydration while it inhibits winter-hemisphere O<sub>3</sub> through hydration. Additionally, the adiabatic cooling of upwelling directly enhances summer-hemisphere O<sub>3</sub> through temperature-dependent ozone kinetics. Following the ozone photochemical equilibrium assumption, the O varies in phase with O<sub>3</sub> and is similarly controlled by summer polar upwelling.
- (3) Temperature modulation. In the upper mesosphere, the total (radiative and chemical) heating of O and O<sub>3</sub> significantly influences the energy budget, creating a positive correlation between O/O<sub>3</sub> and T90. Since upwelling oppositely controls O<sub>3</sub>/O in the two hemispheres, it produces an anti-phase temperature response: T90<sub>S</sub> in the summer polar region shows a negative correlation with T80<sub>S</sub>, while T90<sub>W</sub> in winter hemisphere exhibits a positive correlation with T80<sub>S</sub>.

This theoretical framework enables a quantitative evaluation of variable sensitivities to T80<sub>s</sub> through inter-satellite comparisons, to establish robust causal relationships in the observed climate variability.

**Figure 2.** Schematic diagram of the bottom-up control mechanism in solstice months, showing how summer polar upwelling controls interhemispheric climate variability through coupled dynamical and chemical processes. Key pathways include: (1) winter-hemisphere hydration is induced by summer polar upwelling in combination with meridional transport, and summer-hemisphere dehydration is governed by the cold-trap effect; (2) subsequent modulation of O<sub>3</sub> through H-driven ozone loss and temperature-dependent ozone kinetics, with O varying in pace with O<sub>3</sub> due to the ozone photochemical equilibrium assumption; and (3) resulting temperature (T90) variations via radiative and chemical heating of O<sub>3</sub>/O.

## 3 Results







#### 3.1 Climate patterns in December

Our analysis of MLS/Aura data reveals distinct interannual variability patterns during December (Figure 3). Using T80<sub>S</sub> as an upwelling proxy, we observe opposing H<sub>2</sub>O responses between hemispheres: dehydration in the summer polar mesopause (75°S-82°S) indicated by positive correlations with T80<sub>S</sub> (R=0.93, Fig. 3a), and hydration at low latitudes (15°N-20°N) in the NH indicated by negative correlations (R=-0.84, Fig. 3e). The H<sub>2</sub>O patterns drive corresponding O<sub>3</sub> variations through chemical interaction, with T80<sub>S</sub> showing negative correlations (R=-0.91) with O<sub>3</sub> at 0.001 hPa (65°S-75°S, Fig. 3b) but positive correlations (R=0.94) at 0.002 hPa (15°N-20°N, Fig. 3f). Temperature responses exhibit clear hemispheric antisymmetry, with T90<sub>S</sub> at 0.0046 hPa (65°S-75°S) negatively correlated (R=-0.92, Fig. 3d) and T90<sub>W</sub> at 0.001 hPa (15°N-20°N) positive correlated (R=0.80, Fig. 3h) with T80<sub>S</sub>. In addition, the standard deviations of these variables significantly exceed their linear trends, potentially masking long-term signals.

Figure 4 shows the latitudinal extent of these relationships, with sensitivities derived from linear regression. Meridional transport extends dehydration signature to 35°S (beyond the PMC coverage) and hydration to 35°N (Fig. 4a), while O<sub>3</sub> responses show comparable latitudinal ranges (Fig. 4b). SABER observations corroborate these findings (Figure 5), though atomic hydrogen data gaps at summer high latitudes (Fig. 5a) prevent complete verification of dehydration effects. The ozone photochemical equilibrium assumption holds well, with atomic oxygen O varying in pace with O<sub>3</sub> (Fig. 5b, c). Mlynczak *et al.* (2018) showed that the total (radiative and chemical) heating rate of O and O<sub>3</sub> is ~10 Kday<sup>-1</sup> on global and annual scale (Fig. 3 therein). The relative variations of O and O<sub>3</sub> to T80<sub>S</sub> in Fig. 5 are both ~4%/K, therefore the sensitivity of O/O<sub>3</sub> total heating rate to T80<sub>S</sub> is roughly estimated to be ~0.4 Kday<sup>-1</sup>/K, which partly explain the ~1K/K (Fig. 5d) response of T90 to T80<sub>S</sub>.

Daily-scale SOFIE observations provide further insights (Figures 6-8). Figure 6 shows that O<sub>3</sub> at 90 km is bottom-up controlled by the T80<sub>S</sub> at 79 km. It is noteworthy that in November, during which PMCs are weak and dehydration is absent, the negative correlation between O<sub>3</sub> and T80<sub>S</sub> remains significant, possibly due to the temperature-dependent ozone kinetics. Summer polar O<sub>3</sub> near 0.01 hPa (Fig. 5b) or 80 km (Fig. 7c) is negatively correlated to T80<sub>S</sub>, which should be attributed to ozone kinetics. As shown in Fig. 8b, O<sub>3</sub> below 90 km lags T80<sub>S</sub> by zero days possibly through ozone kinetics, while O<sub>3</sub> above 90 km lags T80<sub>S</sub> by ~3 days, suggesting combined kinetic and dehydration influences. However, for upper-level O<sub>3</sub>, the relative contribution of ozone kinetics and dehydration (depletion of H) is unclear.

Fig. 7a, b shows that the H<sub>2</sub>O above and below PMCs is both bottom-up controlled by T80<sub>S</sub> rather than the temperature above PMC, which supports the cold-trap effect. Fig. 8a further shows that the hydration and dehydration both lag behind T80<sub>S</sub> by about zero days, consistent with the cold-trap effect. In contrast, according to the conventional freeze-drying effect, the dehydration tends to occur prior to the hydration, with the time interval roughly equal to the lifetime of ice particle which is up to two days in simulations (not consider transport and diffusion of ice particles). In addition, T90 is positively

correlated to  $O_3$  (Fig. 7c) and negatively correlated to  $T80_8$  (Fig. 7d), with lag time of about  $\sim 3$  days, matching the radiative and chemical heating of  $O_3/O$ .

## 3.2 Climate patterns in June




The climate patterns in June (Figures 9-12) mirror those of December, but with reduced amplitudes. The T80<sub>s</sub> variability is weaker (1.5 K vs. 5.1 K in December), yielding smaller constituent variations and lower significance (Figs. 3, 9). SABER H responses are insignificant (Fig. 11a), while T90<sub>s</sub> correlations weaken or disappear (Figs. 11d, 12d). MLS data show a localized hydration signal at 0.001 hPa between 70°N~80°N (Fig. 10a), possibly from PMC sublimation due to higher T90<sub>s</sub>.

Interhemispheric transports are more pronounced in June than in December due to differences in gravity wave forcing of the circulation (Smith *et al.*, 2011). The enhanced meridional transport extends H<sub>2</sub>O influences to 40°S~50°S, affecting O<sub>3</sub> and T90<sub>W</sub> in the winter high-latitudes (Fig. 10). Downward winds in the winter polar region generally result in the lowest H<sub>2</sub>O levels near solstice, although vertical winds and diffusion can increase winter polar H<sub>2</sub>O above 90 km (Lossow et al., 2009). Whether the summer polar upwelling in June could influence the winter high-latitude climate through interhemispheric H<sub>2</sub>O transport is an interesting question.

Figure 3. Hemispheric responses of mesospheric (a, e) H<sub>2</sub>O, (b, f) O<sub>3</sub>, and (d, h) T90 to upwelling variability (indicated by T80<sub>S</sub>) during December, showing anti-phase behavior between (left) summer and (right) winter hemispheres. Panels (c, g) show O<sub>3</sub>-T90 correlations. All plots display monthly and zonal means from MLS/Aura observations during 2004-2022, with correlation coefficients (R), sensitivities, standard deviations (σ), and trends derived from linear regression analysis. The opposing H<sub>2</sub>O responses reflect hydration (winter hemisphere, negative correlation) and dehydration (summer hemisphere, positive corelation), while O<sub>3</sub> and T90 variations demonstrate the consequent chemical and thermal feedbacks.

**Figure 4.** Spatial patterns of bottom-up control processes during December showing (a) H<sub>2</sub>O hydration (blue) and dehydration (red) response to upwelling (indicated by T80<sub>S</sub>, white stars), (b) corresponding O<sub>3</sub> variations in both hemispheres, and (c) resultant temperature (T90) changes driven by radiative and chemical heating. Sensitivities of variables to T80<sub>S</sub> derived from linear regression of MLS/Aura data during 2004-2022 show significant anti-phase relationships between hemispheres. Gray points make the 0.95 significance level.

**Figure 5.** Bottom-up control of atmospheric constituents in December from SABER observations during 2002-2019. (a) H response to upwelling (indicated by T80<sub>S</sub>, black stars at 75°S~83°S, 0.01 hPa level). (b) O<sub>3</sub> and (c) O variations, both demonstrating hemispheric anti-symmetry in their sensitivity to T80<sub>S</sub>. (d) Resultant temperature responses showing a positive correlation between T90<sub>W</sub> and T80<sub>S</sub> and a negative correlation between T90<sub>S</sub> and T80<sub>S</sub> through the radiative and chemical heating of O and O<sub>3</sub>. All sensitivities (relative responses in unit of %/K) have been derived from linear regression analysis of monthly data, with consistent upwelling-driven patterns across variables.

Figure 6. Temporal relationship between T80<sub>S</sub> and O<sub>3</sub> variations from SOFIE/AIM observations for the SH PMC seasons during 2007-2013. (Left) Daily time series of T80<sub>S</sub> at 79 km (red) and O<sub>3</sub> at 90 km (black) at ~70°S, with 35-day running mean removed. (Right) Correlation coefficients between T80<sub>S</sub> and O<sub>3</sub> for each PMC season during -10 to 50 days relative to solstice, demonstrating consistent negative relationships (mean R=-0.64) that highlight the bottom-up control of upwelling on ozone chemistry.

Figure 7. Daily-scale bottom-up control processes observed by SOFIE at ~70°S for the SH PMC seasons during 2007-2013.

(a) H<sub>2</sub>O above and below 83 km are both subject to the control of T80<sub>S</sub> rather than local temperatures. (b) Combined H<sub>2</sub>O and ice content demonstrate similar T80<sub>S</sub> dependence. (c) Anti-correlation between O<sub>3</sub> and T80<sub>S</sub>, accompanied by positive O<sub>3</sub>-T90 relationship. (d) Negative correlation between T90<sub>S</sub> and T80<sub>S</sub>. Correlation coefficients are calculated for -10 to 50 days relative to solstice (35-day running mean removed), with the 0.95 significance level marked by dots.

Figure 8. Time-lag analysis of bottom-up control processes from SOFIE observations at ~70°S for the SH PMC seasons during 2007-2013. (a) Immediate response (zero-day lag) of both hydration (negative H<sub>2</sub>O-T80<sub>S</sub> correlation) and dehydration (positive H<sub>2</sub>O-T80<sub>S</sub> correlation) to upwelling. (b) O<sub>3</sub> response showing altitude-dependent lags to T80<sub>S</sub>. (c) T90<sub>S</sub> lags behind O<sub>3</sub> by ~3 days, possibly due to the radiative and chemical heating. (d) T90<sub>S</sub> lags after T80<sub>S</sub> by ~3 days, indicating the cumulative response time of the full upwelling-H<sub>2</sub>O-O<sub>3</sub>-T90<sub>S</sub> chain. Correlation coefficients are calculated similar to Figs. 6-7, with dashed lines marking the 0.95 significance level.

Figure 9. Hemispheric responses of mesospheric (a, e) H<sub>2</sub>O, (b, f) O<sub>3</sub>, (d, h) T90 to upwelling variability (indicated by T80<sub>S</sub>) during June, showing analogous but weaker anti-phase behavior compared to December (Fig. 3). Panels (c, g) show O<sub>3</sub>-T90 correlations. Note the reduced amplitude of T80<sub>S</sub> standard deviation (1.5 K in June vs. 5.1 K in December) and weaker correlations in the June patterns.

Figure 10. Spatial patterns of bottom-up control processes during June showing (a) H<sub>2</sub>O, (b) O<sub>3</sub>, and (c) temperature responses to upwelling (indicated by T80<sub>S</sub>, white stars), analogous to but weaker than December patterns (Fig. 4). Sensitivities derived from linear regression of MLS/Aura data (2004-2022) show reduced but still significant anti-phase relationships between hemispheres compared to December. Gray dots mark the 0.95 significance level. Note the responses of variables in winter hemisphere extend to ~50°S in June.

**Figure 11.** Bottom-up control of atmospheric constituents in June from SABER observations (2002-2019), showing analogous but weaker responses than December (Fig. 5). (a) H shows insignificant response to upwelling (indicated by T80<sub>S</sub>, black stars at 75°N-83°N, 0.006 hPa). (b) O<sub>3</sub> and (c) O variations exhibit reduced hemispheric anti-symmetry in sensitivity to T80<sub>S</sub>. (d) The positive T90<sub>W</sub> responses in wither hemisphere maintain, but the negative T90<sub>S</sub> responses disappear. All sensitivities have been derived from linear regression analysis of monthly data.

**Figure 12.** Daily-scale bottom-up control processes from SOFIE/AIM observations for the NH PMC seasons during 2007-2014. (a) H<sub>2</sub>O in NH shows weaker control by T80<sub>S</sub> compared to that in SH (Fig. 7). (b) Combined H<sub>2</sub>O and ice content demonstrating similar but less pronounced T80<sub>S</sub> dependence. (c) Anti-correlation between O<sub>3</sub> and T80<sub>S</sub> with reduced O<sub>3</sub>-T90<sub>S</sub> relationship. (d) Weaker negative relationship between T90<sub>S</sub> and T80<sub>S</sub>. Correlation coefficients calculated for -10 to 50 days relative to solstice (35-day running mean removed), with black dots marking the 0.95 significance level. Note the generally weaker responses compared to SH observations in Fig. 7.

## 4 Discussion







#### 4.1 H<sub>2</sub>O/H variability mechanisms

Our results demonstrate it is the summer polar upwelling that fundamentally drives the interannual variability of both hydration and dehydration through two distinct pathways: (1) direct upward transport of H<sub>2</sub>O by upwelling combined with meridional wind transport creates hydration in the winter hemisphere; and (2) adiabatic cooling associated with upwelling enhances ice particle growth that blocks upward H<sub>2</sub>O transport, causing dehydration above PMC in the summer hemisphere.

The cold-trap effect differs from the conventional freeze-drying effect in several ways. While freeze-drying effect describes a top-down process dominated by ice particle sedimentation, the cold-trap effect represents a bottom-up process driven primarily by upwelling dynamics. Sedimentation of ice particles plays a dominant role in the freeze-drying effect, however, it is unnecessary in the cold-trap effect (Zhang *et al.*, 2025a). Importantly, the cold-trap effect produces simultaneous hydration and dehydration. In contrast, the freeze-drying effect involves dehydration prior to hydration, with a time lag (estimated as the ice particle lifetime) of up to two days. Particularly, if PMCs are weak or absent and ice particles could not block upward H<sub>2</sub>O transport, hydration could even occur in the absence of dehydration, inconsistent with the freeze-drying effect.

It should be emphasized that we are not attempting to demonstrate that the freeze-drying effect is incorrect. Instead, the two mechanisms should be complementary rather than contradictory. The freeze-drying effect has been verified by simulations for describing PMC microphysical processes at local scales, while the cold-trap effect better explains the global-scale H<sub>2</sub>O redistribution patterns, especially the H<sub>2</sub>O in the winter hemisphere. Additionally, the parameterization of freeze-drying effect is unavailable until now, due to the complexity in PMC simulations. In contrast, the cold-trap effect is much simpler, the interannual variability of hydration/dehydration is only dependent on the T80<sub>S</sub>. The limited H variability detected in SABER data (Figs. 5a, 11a) may reflect both measurement uncertainties in H retrieval and gaps in spatial coverage in high-latitude regions where dehydration effects are strongest.

## 4.2 O<sub>3</sub>/O response pathways

Upwelling forcing drives distinct hemispheric patterns in O<sub>3</sub> variations. In the winter hemisphere, hydration-driven increases in H abundance lead to enhanced O<sub>3</sub> destruction. In the summer hemisphere, however, O<sub>3</sub> increases via two pathways: reduced H abundance due to dehydration and temperature-dependent ozone kinetics caused by adiabatic cooling.

Our vertical analysis shows that these processes operate at different altitudes (Figs. 5b, 7c). Below 85 km, where dehydration is absent, temperature-dependent ozone kinetics dominate O<sub>3</sub> abundance, while above 85 km both dehydration and ozone kinetics contribute, with their relative contribution remaining unclear. In addition, the lag time of O<sub>3</sub> to T80<sub>S</sub> differs for varying altitudes (Fig. 8b), which may be useful for identifying the two pathways.

## 4.3 Temperature modulation processes

The anti-phase temperature responses between hemispheres (T90<sub>W</sub> and T90<sub>S</sub>) reflect the combined effect of dynamical processes and O/O<sub>3</sub> heating. In the winter hemisphere, based on the known radiative and chemical heating rate (~10 K/day on global and annual scale) of O/O<sub>3</sub>, the 3~4%/K sensitivity of O<sub>3</sub> and O in low latitudes to T80<sub>S</sub> (Fig. 5b, c) approximately explains the observed ~0.5 K/K sensitivity of T90<sub>W</sub> to T80<sub>S</sub> (Fig. 5d).

The summer hemisphere presents a more complex scenario involving competing processes. The O<sub>3</sub> concentration in summer polar region is very low due to the destruction by sunlight and H<sub>2</sub>O (Fig. 1b), while the O concentrations do not differ much between the summer and winter hemispheres. The high sensitivity of T90<sub>S</sub> to O<sub>3</sub> (51.7 K/ppmv, Fig. 3c) in December is likely unrealistic. O-heating likely dominates the interannual variability of T90<sub>S</sub>, with O<sub>3</sub>-heating playing a minor role.

Another explanation is that a potential downwelling may result in higher  $T90_S$  by adiabatic heating and higher O concentration by downward transports, which further increase  $O_3$ . The zonal winds change direction above the mesopause height, and gravity waves that induce upwelling near 80 km tend to cause downwelling at altitudes above the mesopause. In this scenario, the positive correlation between  $T90_S$  and  $O/O_3$  results from vertical winds rather than solar or chemical heating. However, Fig. 8c, d shows that the  $T90_S$  lags behind  $O_3$  and  $T80_S$  by  $\sim 3$  days, ruling out pure dynamical explanations. Fig. 4c and Fig. 5d shows that the  $T90_S$  at latitudes as low as  $30^\circ S$  are negatively correlated with  $T80_S$ , which should not be attributed to the adiabatic heating of downwelling. Moreover, shrinking effect may explain the negative correlation between  $T90_S$  and  $T80_S$  through the vertical shift of the temperature profile. However, the MLS and SABER results are presented on pressure levels rather than geometric heights, not supporting a shrinking effect as the major driver for the interannual variability of  $T90_S$ . In addition,  $T90_S$  lags behind  $T80_S$  by  $\sim 3$  days (Fig. 8d), inconsistent with the shrinking effect on the daily scale.

## 310 5 Conclusion






This study establishes that summer polar upwelling serves as the primary driver of interannual variability in the upper mesospheric  $H_2O(H)$ ,  $O_3(O)$ , and T90 through a bottom-up mechanism. Our findings build a simple picture for understanding the interhemispheric anti-phase climate of the upper mesosphere, revealing three key aspects of dynamical-chemical interactions: First, we demonstrate that the summer polar upwelling drives the global-scale interannual  $H_2O$  variability, namely the hydration in the winter hemisphere and dehydration in the summer hemisphere. Second, owing to the negative modulation of  $O_3$  by  $H_2O$  and the ozone photochemical equilibrium assumption, the winter-hemisphere  $O_3/O$  decreases due to the hydration-induced depletion of H, while the summer-hemisphere  $O_3/O$  increases due to combined effects of dehydration and temperature-dependent ozone kinetics. Third, because of the radiative and chemical heating of  $O_3/O$ , there is a positive correlation between the winter-hemisphere  $O_3/O$  and  $O_3/O$ , there is a positive correlation between the winter-hemisphere  $O_3/O$  and  $O_3/O$ , and  $O_3/O$  and

Several important questions remain unresolved and need further investigation. Summer polar upwelling plays a pivotal role in the bottom-up control mechanism, while the pronounced hemispheric difference in the interannual T80<sub>s</sub> variability (5.1 K in December, 1.5 K in June) lacks a detailed explanation. Given the importance of H<sub>2</sub>O for mesospheric climate, the relationship between our proposed cold-trap effect and conventional, well-established freeze-drying effect should be a priority for further study. Similarly, the relative contribution of dehydration versus ozone kinetics to summer-hemisphere O<sub>3</sub> enhancement require quantification. Alternative explanations for the interannual T90<sub>s</sub> variability, including shrinking effect and adiabatic heating of gravity wave-driven downwelling, cannot fully account for the observed patterns, but may make some contribution. These advances will further elucidate mesospheric climate variability while improving the physical basis for interpreting long-term trends in this intricate atmospheric region.


325

Data Availability. The MLS Level 2, version 5 data are available for download from the NASA GES DISC website. Specifically, the MLS water vapor data can be accessed at https://disc.gsfc.nasa.gov/datasets/ML2H2O 005/summary, the https://disc.gsfc.nasa.gov/datasets/ML2O3 005/summary, ozone data at and the data temperature at https://disc.gsfc.nasa.gov/datasets/ML2T 005/summary. The quality document of MLS available https://mls.jpl.nasa.gov/data/v5-0 data quality document.pdf. The SABER data version 2.07 of temperature, ozone, atomic oxygen, and atomic hydrogen can be download from https://data.gats-inc.com/saber/Version2 0/SABER atox/. The PMC 340 data from SOFIE/AIM are available from the SOFIE website (http://sofie.gats-inc.com/sofie/index.php).

Author Contributions. LZ, ZL, and BT conceived the idea together. LZ analyzed the data and drafted the manuscript. ZL and BT revised the paper.

Competing Interests. The authors declare that they have no conflict of interests.

Acknowledgements. We are especially grateful to the MLS, SABER, and SOFIE program for providing us with the long-range and high-quality data.

Financial support. This research has been supported the National Natural Science Foundation of China (42025602 and 41905059).

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
