# Peer review of "Interhemispheric Anti-Phase Variability in Mesospheric Climate Driven by Summer Polar Upwelling During Solstice Months"

_EGUsphere, 2025_

## Referee Comment (RC1)

The manuscript investigates chemical and thermal variability in the mesosphere, with a particular focus on the role of upwelling in the polar summer. The study is based on satellite data (MLS, SABER, SOHIE). A major finding of the manuscript is a causal chain involving upwelling, transport of water vapour and resulting odd hydrogen, effects of odd hydrogen on odd oxygen, and the role of odd oxygen on heating the upper mesosphere. In polar summer condition, the additional effect of Polar Mesospheric Clouds on water vapour is considered (cold trap, dehydration above the cloud).

Relationships between relevant quantities are investigated using correlation analysis and time-lag analysis applied to the satellite data. The temperature around 80 km altitude is used as a proxy for the strength of the mesospheric upwelling, which makes sense considering the adiabatic cooling connected to the upwelling.

Model simulations would be beneficial for this study, as they could help to distinguish between the role of various processes. However, I understand that model simulations are beyond the scope of the current manuscript. Rather, the major achievement of this manuscript is tracing the suggested mechanism in the satellite data.

The manuscript is well written. The figures are very well designed and instructive.

I recommend the manuscript for publication after some major revisions and text corrections.

General comments:

The manuscript is instructive and nicely illustrates the suggested relationships. However, I disagree with the authors in calling this a "novel mechanism". Rather the individual processes and relationships described in the manuscript are well known and already part of atmospheric numerical model. This also includes the effect of the mesospheric cold trap for models that include PMC microphysics or that are coupled to PMC microphysics. As stated above, the manuscript is valuable in clearly describing the connections between upwelling, odd hydrogen abundance, odd oxygen abundance, and heating. But do not call it a novel mechanism.

The manuscript correctly describes that upwelling affects odd oxygen through (1) lower temperatures that affect the reaction $O + O_2 + M$, and (2) generation of PMC, resulting in dehydration and reduced $HO_x/O_x$ chemistry. However, there is another major effect of upwelling/downwelling on odd oxygen concentrations in the upper mesosphere: controlling the transport from the large atomic oxygen reservoir in the lower thermosphere. The authors address this process late in the manuscript (Section 4.3, Lines 294-300). Based on the time lag analysis, the authors rule out a major effect of this transport on the observed variability of odd oxygen in the upper mesosphere. Nonetheless, the variability of odd oxygen sources is a very important topic for this manuscript. Therefore, I suggest to describe sources of odd oxygen and their variability already in Section 1.2 ("Ozone chemistry and variability"). Currently, this section only focuses on sinks of odd oxygen.

The authors state that there is a fundamental difference between the "freeze-drying effect" by PMCs as described in earlier literature and the "cold trap" effect by PMCs described in the current manuscript. I do not agree. Both are based on the same physical principle: water vapour freezes to PMC particles while the PMC particles sediment, the particles then sublimate and release the water vapour. What varies from case to case is the altitude difference (and thus the time difference) between the freezing and the sublimation. The authors may want to argue that altitude difference may be smaller than suggested by other studies. However, they should not claim that this a fundamentally different process ("cold trap" rather than "freeze drying").

Earlier studies of the PMC cold trap generally found a local or a regional effect: The cold trap affects the distribution of water vapour in the polar mesosphere region. The current manuscript suggests that the PMC cold trap affects water vapour all the way to lower latitudes and to the winter hemisphere. I do not find this convincing, and I do not think that the PMC cold trap is needed to explain enhanced water vapour ("hydration") at these latitudes. Rather, a much more direct explanation of the connection between T80s (upwelling) and the global hydration in Figure 4a and Figure 10a is the continuity equation: When there is more upwelling near the summer pole, mass conservation will lead to horizontal transport of water vapour towards lower latitudes and the winter hemisphere.

Hence, some statements must be revised. Two examples: Line 137, "cold-trap effect induced winter hemisphere hydration" should be replaced with "winter hemisphere hydration that is induced by summer pole upwelling in combination with meridional transport". Line 309-310, "we demonstrate that the cold-trap effect effectively explains global-scale interannual $H_2O$ variability" should be replaced by "we demonstrate that updraft in combination with meridional transport effectively explains global-scale interannual $H_2O$ variability"

Section 1 provides a good review of the processes connected to water, PMC, odd oxygen and the thermal balance in the mesosphere. Much of Section 2.2 is then a repetition of Section 1. I recommend to shorten Section 2.2.

Text corrections:

Line 9: Remove "the" from "Using the MLS..."

Line 10: The manuscript refers to temperatures and concentration at specific altitudes, not in altitude ranges. Therefore, clarify by replacing "above 90 km" with "near 90 km".

Line 13: Change to "toward the winter hemisphere".

Line 14-15: Change "... due to ozone chemical equilibrium assumption, and..." to "... due to chemical equilibrium, and..."

Line 15: Change "chemical heating of O/O3 reduces the T90 in winter hemisphere..." to "chemical heating by O/O3 reactions reduces T90 in the winter hemisphere..."

Line 33: Clarify by replacing "... $H_2O$ maxima in December" by "... $H_2O$ maxima in the summer hemisphere"

Line 39: Change to "suggesting an incomplete understanding"

Line 32: Solar UV is expected to affect H2O, which in turn is experctyed to affct PMC. Therefore, clarify by changing to "influences H2O, and thus PMCs, (Rehmberg et al..."

Line 54: Clarify by changing to "...these advances in understanding, ..."

Line 66: Change to "with a rate exceeding..."

Line 69: Change to "exhibiting an unexpected warming"

Line 73: Change to "affects mesospheric winds"

Line 91-92: Clarify by changing to "...orbits, focusing on zonal..."

Line 102: Change "instruments" to "instrument".

Line 107: Change to "after which SOFIE measurements shifted..."

Line 110: Change to "upwelling on interannual timescales".

Line 119: Change to "towards the winter hemisphere".

Line 123: Change to "while it inhibits..."

Line 130: Change to "T90s in the summer..."

Line 132: Change to "enables a quantitative..."

Line 139: Change to "to the ozone..."

Line 162: Change to "that $O_3$ at 90 km is..."

Line 163: Change to "during which PMCs are weak..."

Line 206: Change to "... %/K) have been derived from..."

Line 226: Change tp "similar to Figs. 6-7..."

Line 245: Change to "... sensitivities have been derived from..."

Line 280: Change to "shows that these processes..."

Line 295-296: Change to "above the mesopause height. and gravity waves..."

Line 296: Change "tends" to "tend".

Line 296: Change to "at altitudes above the mesopause"

Line 301: Change to "or the temperature profile"

Line 302: Change to "supporting a shrinking effect"

Line 308: Change to "climate of the upper mesosphere"

Line 312: Change "depletion of H" to "depletion by H"

---

## Author Comment (AC1)

**Response to Referees' Comments on egusphere-2025-2047, "Interhemispheric Anti-Phase Variability in Mesospheric Climate Driven by Summer Polar Upwelling During Solstice Months"**

We sincerely appreciate the time and effort the two Referees have taken to review our work. Their comments are valuable and constructive, and help us to improve the quality of our manuscript. The following are the point-by-point responses.
* * *
Response to Referee #1
* * *
The manuscript investigates chemical and thermal variability in the mesosphere, with a particular focus on the role of upwelling in the polar summer. The study is based on satellite data (MLS, SABER, SOHIE). A major finding of the manuscript is a causal chain involving upwelling, transport of water vapour and resulting odd hydrogen, effects of odd hydrogen on odd oxygen, and the role of odd oxygen on heating the upper mesosphere. In polar summer condition, the additional effect of Polar Mesospheric Clouds on water vapour is considered (cold trap, dehydration above the cloud). Relationships between relevant quantities are investigated using correlation analysis and time-lag analysis applied to the satellite data. The temperature around 80 km altitude is used as a proxy for the strength of the mesospheric upwelling, which makes sense considering the adiabatic cooling connected to the upwelling.

We are grateful for the reviewer's thorough and accurate summary of our manuscript's scope and key findings.

Model simulations would be beneficial for this study, as they could help to distinguish between the role of various processes. However, I understand that model simulations are beyond the scope of the current manuscript. Rather, the major achievement of this manuscript is tracing the suggested mechanism in the satellite data.

We thank the reviewer for suggesting the value of model simulations. Although the primary focus of this study is to establish observational evidences for the proposed coupling mechanisms using satellite data, this suggestion is an important direction for future work. We fully agree that complementary modeling work could help disentangle the relative contributions of different mechanisms in this work.

The manuscript is well written. The figures are very well designed and instructive. I recommend the manuscript for publication after some major revisions and text corrections.

We appreciate the reviewer's positive comments on the language and figure design. The major and editorial comments have been responded in the following.

**General comments:**

The manuscript is instructive and nicely illustrates the suggested relationships. However, I disagree with the authors in calling this a "novel mechanism". Rather the individual processes and relationships described in the manuscript are well known and already part of atmospheric numerical model. This also includes the effect of the mesospheric cold trap for models that include PMC microphysics or that are coupled to PMC microphysics. As stated above, the manuscript is valuable in clearly describing the

connections between upwelling, odd hydrogen abundance, odd oxygen abundance, and heating. But do not call it a novel mechanism.

> We agree that individual relationships (e.g., upwelling—$H_2O$, $H_2O$—$O_3/O$, $O_3/O$—T) are well-established, as shown in the literatures cited by our manuscript.
> In Line 78, "This paper establishes a novel bottom-up mechanism…" has been changed to "This paper elucidates a systematic bottom-up mechanism…", in order to avoid claiming them as a novel mechanism, and emphasize our observational synthesis of the known processes and their coupled effects.

The manuscript correctly describes that upwelling affects odd oxygen through (1) lower temperatures that affect the reaction $O + O_2 + M$, and (2) generation of PMC, resulting in dehydration and reduced $HO_x/O_x$ chemistry. However, there is another major effect of upwelling/downwelling on odd oxygen concentrations in the upper mesosphere: controlling the transport from the large atomic oxygen reservoir in the lower thermosphere. The authors address this process late in the manuscript (Section 4.3, Lines 294-300). Based on the time lag analysis, the authors rule out a major effect of this transport on the observed variability of odd oxygen in the upper mesosphere. Nonetheless, the variability of odd oxygen sources is a very important topic for this manuscript. Therefore, I suggest to describe sources of odd oxygen and their variability already in Section 1.2 ("Ozone chemistry and variability"). Currently, this section only focuses on sinks of odd oxygen.

> We thank the reviewer for this suggestion.
> In Line 46-48, a detailed introduction of the lifetime and variability of odd oxygen is presented, by citing the work of *Smith et al.* (2010), and the role of vertical transports is highlighted, providing better context for the later analysis in section 4.3.

The authors state that there is a fundamental difference between the "freeze-drying effect" by PMCs as described in earlier literature and the "cold trap" effect by PMCs described in the current manuscript. I do not agree. Both are based on the same physical principle: water vapour freezes to PMC particles while the PMC particles sediment, the particles then sublimate and release the water vapour. What varies from case to case is the altitude difference (and thus the time difference) between the freezing and the sublimation. The authors may want to argue that altitude difference may be smaller than suggested by other studies. However, they should not claim that this a fundamentally different process ("cold trap" rather than "freeze drying").

> We sincerely thank the reviewer for raising this critical perspective.
> As noted by the reviewer, the role of ice sedimentation is indeed crucial. In our companion manuscript submitted to ACP (Zhang *et al.*, 2025), we proposes an alternative ice particle formation mechanism—the Charged Meteoric smoke particle Nucleation (CMN) scheme—which differs from the conventional growth-sedimentation (GS) scheme. The CMN scheme is based on two assumptions: nucleation occurs throughout PMC altitude range with charged meteoric smoke particle acting as ice nuclei, and ice particles grows predominantly in situ. In particular, the sedimentation of ice particle is negligible in the CMN scenario, providing a microphysical basis for the cold-trap effect.
> Importantly, our proposed CMN scheme and the resulting cold-trap effect do not conflict with the GS scheme and freeze-drying effect, as they may operate at different scales (zonal and daily/interannual vs. individual PMCs).

> Regardless the microphysical interpretations, our current study demonstrates that upwelling unequivocally drives the observed dehydration above PMCs. While the detailed comparison between cold-trap and freeze-drying effects is beyond the scope of this paper, we maintain that they represent distinct microphysical pathways.

Earlier studies of the PMC cold trap generally found a local or a regional effect: The cold trap affects the distribution of water vapour in the polar mesosphere region. The current manuscript suggests that the PMC cold trap affects water vapour all the way to lower latitudes and to the winter hemisphere. I do not find this convincing, and I do not think that the PMC cold trap is needed to explain enhanced water vapour ("hydration") at these latitudes. Rather, a much more direct explanation of the connection between T80s (upwelling) and the global hydration in Figure 4a and Figure 10a is the continuity equation: When there is more upwelling near the summer pole, mass conservation will lead to horizontal transport of water vapour towards lower latitudes and the winter hemisphere.

> We fully agree that the winter-hemisphere hydration originates from the combined transports of summer polar upwelling and meridional winds. In contrast, the cold-trap effect or freeze-drying effect is mainly responsible for the local dehydration above PMCs.
> We appreciate the reviewer's insightful clarification regarding the dehydration and hydration processes, which significantly improving the physical precision of our mechanism.

Hence, some statements must be revised. Two examples: Line 137, "cold-trap effect induced winter hemisphere hydration" should be replaced with "winter hemisphere hydration that is induced by summer pole upwelling in combination with meridional transport". Line 309-310, "we demonstrate that the cold-trap effect effectively explains global-scale interannual $H_2O$ variability" should be replaced by "we demonstrate that updraft in combination with meridional transport effectively explains global-scale interannual $H_2O$ variability"

> Done. Thanks.
> In Line 137 and Lines 309-310, we have revised all relevant statements as suggested. In addition, in Figure 4, the text "cold-trap effect" has been replaced by "upwelling".

Section 1 provides a good review of the processes connected to water, PMC, odd oxygen and the thermal balance in the mesosphere. Much of Section 2.2 is then a repetition of Section 1. I recommend to shorten Section 2.2.

> Agreed. Thanks.
> Section 2.2 has been shortened by eliminating redundancy.

**Text corrections:**
Line 9: Remove "the" from "Using the MLS...".
Line 10: The manuscript refers to temperatures and concentration at specific altitudes, not in altitude ranges. Therefore, clarify by replacing "above 90 km" with "near 90 km".
Line 13: Change to "toward the winter hemisphere".
Line 14-15: Change "... due to ozone chemical equilibrium assumption, and..." to "... due to chemical equilibrium, and..."
Line 15: Change "chemical heating of $O/O_3$ reduces the T90 in winter hemisphere..." to "chemical heating by $O/O_3$ reactions reduces T90 in the winter hemisphere..."

Line 33: Clarify by replacing "... H$_2$O maxima in December" by "... H$_2$O maxima in the summer hemisphere"

Line 39: Change to "suggesting an incomplete understanding"

Line 42: Solar UV is expected to affect H$_2$O, which in turn is expected to affect PMC. Therefore, clarify by changing to "influences H$_2$O, and thus PMCs, (Remsberg et al..."

Line 54: Clarify by changing to "...these advances in understanding, ..."

Line 66: Change to "with a rate exceeding..."

Line 69: Change to "exhibiting an unexpected warming"

Line 73: Change to "affects mesospheric winds"

Line 91-92: Clarify by changing to "...orbits, focusing on zonal..."

Line 102: Change "instruments" to "instrument".

Line 107: Change to "after which SOFIE measurements shifted..."

Line 110: Change to "upwelling on interannual timescales".

Line 119: Change to "towards the winter hemisphere".

Line 123: Change to "while it inhibits..."

Line 130: Change to "T90s in the summer..."

Line 132: Change to "enables a quantitative..."

Line 139: Change to "to the ozone..."

Line 162: Change to "that O$_3$ at 90 km is..."

Line 163: Change to "during which PMCs are weak..."

Line 206: Change to "... %/K) have been derived from..."

Line 226: Change to "similar to Figs. 6-7..."

Line 245: Change to "... sensitivities have been derived from..."

Line 280: Change to "shows that these processes..."

Line 295-296: Change to "above the mesopause height, and gravity waves..."

Line 296: Change "tends" to "tend".

Line 296: Change to "at altitudes above the mesopause"

Line 301: Change to "of the temperature profile"

Line 302: Change to "supporting a shrinking effect"

Line 308: Change to "climate of the upper mesosphere"

Line 312: Change "depletion of H" to "depletion by H"

Done. Thanks. All above editorial comments have been addressed.
* * *
Response to Referee #2
* * *
I consider that this paper presents an important contribution to mesospheric climate. It identifies a dynamical and chemical coupling mechanism that drives interhemispheric variability during solstice months. Using satellite datasets the authors assess how the summer polar upwelling affects H$_2$O and O$_3$ chemistry in both hemispheres.

We are grateful for the reviewer's positive evaluations of our work, which are really encouraging for us.

They identify what they call the "cold-trap effect" and distinguish it from the "freeze-drying effect", which I find very interesting.

> Thanks for this comment.
>
> It should be noted that the cold-trap effect acts at zonal and daily/interannual scales, and in fact we did not challenge the freeze-drying effect in this paper.
>
> We have submitted another paper to ACP (Zhang *et al.*, 2025), which proposes an alternative formation mechanism for ice particles in polar mesospheric clouds (PMCs), providing a foundation for the cold-trap effect. Explicitly, the sedimentation of ice particles is assumed to be negligible, allowing the bottom-up modulation of $H_2O$ by PMCs via the cold-trap effect.

The manuscript is well explained, in my opinion, based on the data analysis presented, with very clear figures and data analysis. I have only a couple of question, which does not need to be addressed in the manuscript itself, as I consider the work suitable for publication.

(1) Can the results proposed in the paper be supported by atmospheric models such as WACCM-X for example?

> We appreciate the reviewer's suggestion regarding the potential use of WACCM-X to support our findings. The current study is primarily an observational analysis based on satellite datasets, and incorporating model simulations would exceed the scope of this paper.
>
> The WACCM-X is certainly valuable for better understandings the observations, however, the incorporation of PMCs' impacts on mesospheric chemical environments into models may be a difficult task. The benchmark PMC models (e.g., CARMA) simulate PMC lifecycles by tracing the trajectories of each ice particles (Rapp and Thomas, 2006), which would be time-consuming on zonal scales. Particularly, it is unclear whether the bottom-up cold trap effect could be well depicted, as current PMC models are based on the top-down growth-sedimentation scheme (Hultgren and Gumbel, 2014).
>
> We are trying to develop a simplified 0-D or 1-D PMC model based on an under-review work (Zhang *et al.*, 2025), in which the sedimentation of ice particles is negligible and tracing their trajectories becomes unnecessary. If such a simplified PMC model could be built and incorporated into atmospheric models, the findings in this paper could be further checked. We believe the comparisons between models and observations will be an important next step.

(2) Regarding the observed interannual variability, is there a possibility that global circulation patterns, like the QBO, influence each hemisphere, during solstices, inversely at these altitudes?

> We thank the reviewer for highlighting the importance of QBO in mesospheric climate.
>
> On the one hand, we think the direct effect of QBO may be limited. In this paper, summer polar upwelling near 80 km controls the mesospheric climate during solstices, and in turn is modulated by stratospheric winds via the filtering of gravity waves. However, according to the Holton-Tan effect, the influences of QBO toward high latitudes are significant in the winter hemisphere but insignificant in the summer hemisphere (Anstey *et al.*, 2022). As a result, the QBO is unlikely to affect the summer polar upwelling through modulating summer-hemisphere stratospheric circulations. In fact, figure 3 reveals that the dominant $T80_S$ periods (3~4 years) is distinct from typical QBO timescales (~28 months).
>
> On the other hand, the QBO may still be important for mesospheric climate, but through other pathways. For example, the QBO's influence could extend into the winter stratosphere via the

Holton-Tan effect, which further affects mesospheric climate through the interhemispheric coupling (Karlsson *et al.*, 2007); the mesospheric QBO and semiannual oscillation are correlated with the stratospheric QBO, possibly affecting low-latitude mesosphere.

**References:**

Anstey, J. A., S. M. Osprey, J. Alexander, M. P. Baldwin, N. Butchart, L. Gray, Y. Kawatani, P. A. Newman, and J. H. Richter (2022), Impacts, processes and projections of the quasi-biennial oscillation, *Nature Reviews Earth & Environment*, *3*(9), 588-603, doi:https://doi.org/10.1038/s43017-022-00323-7.

Hultgren, K., and J. Gumbel (2014), Tomographic and spectral views on the lifecycle of polar mesospheric clouds from Odin/OSIRIS, *Journal of Geophysical Research: Atmospheres*, *119*(24), 14,129-114,143, doi:https://doi.org/10.1002/2014JD022435.

Karlsson, B., H. Körnich, and J. Gumbel (2007), Evidence for interhemispheric stratosphere-mesosphere coupling derived from noctilucent cloud properties, *Geophysical research letters*, *34*(16), doi:https://doi.org/10.1029/2007GL030282.

Rapp, M., and G. E. Thomas (2006), Modeling the microphysics of mesospheric ice particles: Assessment of current capabilities and basic sensitivities, *Journal of Atmospheric and Solar-Terrestrial Physics*, *68*(7), 715-744, doi:https://doi.org/10.1016/j.jastp.2005.10.015.

Smith, A. K., D. R. Marsh, M. G. Mlynczak, and J. C. Mast (2010), Temporal variations of atomic oxygen in the upper mesosphere from SABER, *Journal of Geophysical Research: Atmospheres*, *115*(D18), doi:https://doi.org/10.1029/2009JD013434.

Zhang, L., Z. Liu, and B. Tinsley (2025), Altitude-Dependent Formation of Polar Mesospheric Clouds: Charged Nucleation and In Situ Ice Growth on Zonal and Daily Scales, *EGUsphere*, *2025*, 1-29, doi:https://doi.org/10.5194/egusphere-2025-2330.

---

## Author Response (AR2)

**Response to comments of referee #2 on egusphere-2025-2047, "Interhemispheric Anti-Phase Variability in Mesospheric Climate Driven by Summer Polar Upwelling During Solstice Months"**

\_\_\_\_\_\_

I would like to thank the authors for thorough responses to the referee comments and for the revised manuscript. I am generally happy with the revisions and I find the manuscript almost ready for publication.

We sincerely thank Referee #2 for the positive feedback on our manuscript. We appreciate the constructive comments, which are invaluable in improving the quality of this work.

I am still uneasy regarding the authors' distinction between freeze-drying (involving vertical transport of PMC particles) and cold trap (not involving vertical transport of PMC particles). I understand that there is a "companion manuscript" by Zhang et al. under review at ACP. It seems that that review process is almost completed, and I suggest to refer to the companion manuscript (discussion paper) in the present manuscript. Once the companion paper will be published, I assume that future studies will need to address the suggested distinction and the relative importance of the PMC freeze drying and PMC cold trap mechanisms. In any case, the focus of the present manuscript is on the role of upwelling for the mesospheric chemistry, and I would argue that the conclusions of the present manuscript are actually not critically dependent on the distinction between freeze-drying and cold trap. Maybe the authors want to comment on that. So, as of now, I am satisfied with the description of both processes in the current manuscript.

We thank the referee for raising this important point regarding the distinction between the freeze-drying and cold-trap effects.

We have now cited our companion manuscript (Zhang et al., 2025) *in lines 124 and 268*, which provides detailed microphysical foundations supporting the cold-trap effect.

We fully agree that the conclusions of this paper—specifically concerning the role of upwelling in mesospheric chemistry—do not critically depend on the precise distinction between these two effects. Both mechanisms result in to similar patterns of dehydration above and hydration below PMCs, which adequately explain the chemical responses presented in this study.

However, there is one related issue that must be clarified in order to avoid potential confusion for readers not familiar with water vapour in the middle atmosphere:

The main reason why the upper mesosphere and thermosphere are dry ("dehydrated") is the photolysis of water vapour by solar UV radiation. This is particularly prominent in the polar summer mesosphere because of the permanent solar irradiation. This basic fact should be clearly stated in the manuscript when mesospheric water vapour is discussed (sections 1.1, 2.2 (point 1), and 4.1).

In the polar summer, PMC add to this dehydration of the upper mesosphere and thermosphere, but they are not the major cause. The manuscript uses wordings like "ice particle growth that blocks upward H2O transport, causing de-hydration above PMC in the summer hemisphere" (line 259-260). Such wordings are potentially confusing as a reader may understand that PMC are the major cause of low water concentrations at higher altitudes.

An instructive reference is e.g. Siskind et al. (2018) "Understanding the effects of polar mesospheric clouds on the environment of the upper mesosphere and lower thermosphere". This paper is already cited in the present manuscript, albeit only as an example of PMC effecting odd hydrogen chemistry.

The more important message from that paper is how much PMC actually affect the global water vapour budget water in the mesosphere (based on WACCM model simulations). See in particular their Figure 1, comparing the global distribution of water vapour in a world with PMC and without PMC. I suggest to add this to the discussion with an explicitly reference to this paper.

We sincerely thank the referee for this important clarification.

We agree that it is essential to clearly emphasize the dominant role of solar UV photolysis in dehydrating the upper mesosphere, particularly in the polar summer region under continuous illumination. This fundamental mechanism has now been explicitly highlighted *in lines 32-33* of the revised manuscript.

As noted by the referee, although PMCs contribute to local dehydration above the cloud layer, they are a secondary factor compared to photolytic loss. To avoid potential misinterpretation, we have clarified *in line 261* that the PMC-related dehydration/hydration primarily concern the interannual variability in water vapor, rather than the background water vapor profile which is dominated by solar UV photolysis.

We also fully agree on the relevance of Siskind et al. (2018) and thank the referee for underscoring its importance. Given that most WACCM simulations do not include an interactive PCM module, the results presented by Siskind et al. (2018) offer particularly valuable model-based insight into PMC-induced water vapor redistribution—especially through the comparative analysis (Figure 1 therein) of simulations with and without PMC feedback. *In lines 39-40*, we have expanded our discussion to explicitly reference their findings.